# An Improved Spread-Spectrum Technique for Reduction of Electromagnetic Emissions of Wireless Power Transfer Systems

Deniss Stepins [1,*], Dhruv Deveshkumar Shah [2], Aleksandrs Sokolovs [1] and Janis Zakis [1]

1 Institute of Industrial Electronics and Electrical Engineering, Riga Technical University, 12/1 Azenes Street, 1658 Riga, Latvia
2 Institute of Radio Electronics, Riga Technical University, 12 Azenes Street, 1658 Riga, Latvia
* Correspondence: deniss.stepins@rtu.lv

**Abstract:** The application of conventional spread spectrum techniques for conducted electromagnetic emission (EME) reduction in inductive-resonant wireless power transfer (WPT) systems may not reduce conducted EME enough due to specific frequency characteristics of the resonant systems and it can lead to some "adverse effects", mainly in terms of decreased efficiency. Therefore, in this paper, an improved spread spectrum approach, multi-switching frequency and multi-duty cycle (MFMD) technique, is proposed. The proposed approach can give a considerably better conducted EME reduction along with a better efficiency than the conventional spread spectrum techniques based on a multi-switching frequency scheme. In the proposed approach, the inductive-resonant WPT system can operate at multiple switching frequencies (e.g., three different frequencies) and for a part of a control signal with a specific switching frequency, there is a specific duty cycle. The technique can be implemented in a simple way using an inexpensive 8-bit microcontroller. The effect of the MFMD scheme on the conducted EME and efficiency of the WPT system is studied in detail. The WPT system conducted EME and the efficiency are studied experimentally with a designed laboratory prototype. The performance characteristics of the WPT system with the MFMD scheme are compared to those with the multi-switching frequency scheme and without the spread spectrum. The WPT system with the proposed spread spectrum technique has a better performance than that with the conventional spread spectrum technique.

**Keywords:** wireless power transfer; inductive-resonant; spread spectrum; conducted emissions; efficiency





## 1. Introduction

With the enormous development of different electronic and electrical devices, wireless power transfer (WPT) has become a very popular topic of research nowadays, because WPT is a more reliable and convenient approach for power transmission than traditional power transmission with wires. WPT is especially important when it is necessary to charge batteries dynamically (while moving) or charge batteries statically when a charging object operates in an automatic mode (such as mobile robots in warehouses) [1–4]. In fact, the static charging of the batteries of mobile robots has attracted a remarkable amount of attention from both the research and the engineering communities, due to the absence of sparking (it is especially important if mobile robots operate at chemical plants), high reliability (no problems with mobile robots' contacts wearing down), no need for very accurate positioning (some misalignment of the WPT coils does not lead to a disruption of the charging process), etc. [1,5].

Although different approaches for wireless power transmission exist, in applications, such as the static charging of the batteries of mobile robots, an inductive-resonant approach is the most useful, as it has a high enough efficiency for small-gap applications [6] when electric power should be transferred wirelessly over a distance up to some tens of cm. Usually, the inductive-resonant WPT systems are designed to operate within the range of

kHz (e.g., within the frequency ranges of 80–90 kHz or 100–200 kHz); however, sometimes they can also be designed to operate within the range of MHz (e.g., at 6.78 MHz) to achieve a higher transmission distance, better spatial freedom and a smaller size.

Due to the presence of switch mode power converters inside, the inductive-resonant WPT systems are the potential sources of electromagnetic emissions (EME) which can cause electromagnetic interference (EMI) to sensitive electronic devices, such as mobile phones [7]. As it is shown in Figure 1, EME can take the form of conducted EME propagating through input wires to an electric grid and radiated EME (time-varying magnetic fields or even radio waves), which can disrupt the normal operation of sensitive electronic devices. Therefore, the EME must be reduced. This paper is devoted to the reduction of conducted EME.

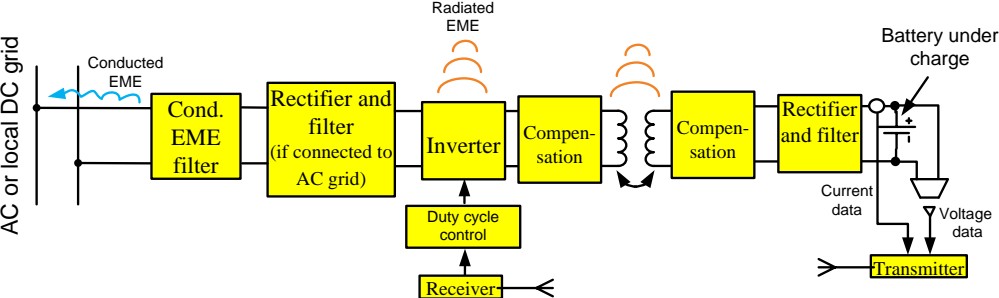

**Figure 1.** Simplified block diagram of the inductive-resonant WPT system connected to a grid irradiating EME.

WPT systems are often considered as industrial, scientific and medical (ISM) devices. Therefore, they should comply with the requirements of the CISPR 11 standard [8]. As it is shown in Figure 1, WPT systems can be connected to either utility AC or local DC electric grids. With an ever-growing interest in renewable energy, DC microgrids have become widespread in both residential and industrial sectors. Therefore, WPT systems can often be connected to DC microgrids and make the EMI conducted to sensitive electronic devices connected to the same grid [9]. As an example, the maximum allowable conducted EME limits for Group 1 industrial electrical devices connected to a DC grid are shown in Figure 2.

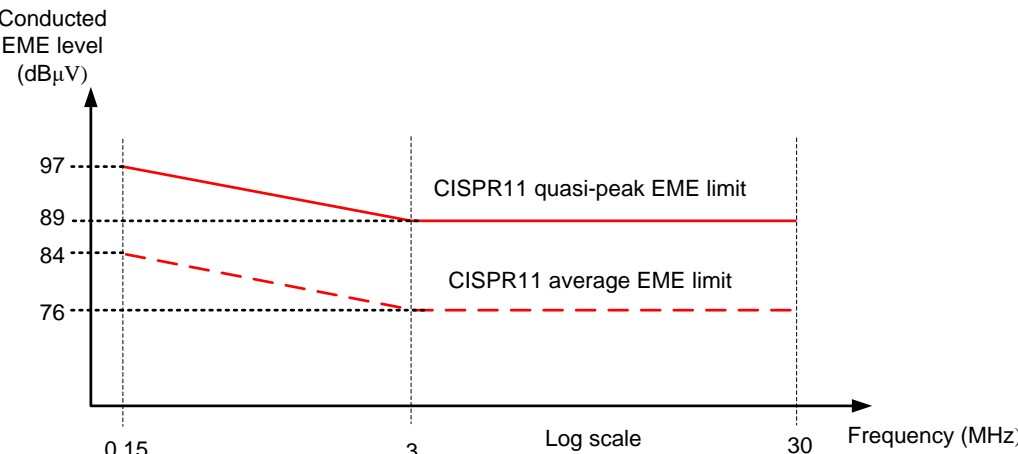

**Figure 2.** Group 1 Class A (industrial devices) CISPR11 limits of conducted EME of a device connected to a local DC grid. Note: the modified figure is copied from [8] and reprinted with the permission from the Academy of EMC.

A classical approach for conducted EME reduction—input conducted EME filters—adds noticeable cost, size and weight to the inductive-resonant WPT system. Therefore, a spread spectrum technique based on WPT inverter switching frequency modulation

has been applied to the inductive-resonant WPT systems [10–13]. A typical approach for the reduction of radiated EME is electromagnetic shielding, but the shields are quite expensive and they increase the size and cost of the WPT systems. In contrast to filtering and shielding, the spread spectrum approach does not require increases in size, cost or weight of WPT systems, because it can be implemented by using a microcontroller with a suitable program code.

There are different papers related to the suppression of conducted EME using the spread spectrum approach in the inductive-resonant WPT systems [10–13] which show that the spread spectrum technique along with moderate conducted EME reduction can lead to some "adverse effects", mainly in terms of the decrease in efficiency. Moreover, as pointed out in [13], the conducted EME reduction coefficient is not high because of the uneven distribution of the conducted EME energy within the first harmonic sidebands, especially for higher switching frequency deviations. Thus, it is of great importance to propose an improved spread spectrum approach that can give a noticeably better conducted EME reduction along with a better efficiency. Therefore, the novelty of this paper (this paper is partly based on the results of a Master's thesis [14]) is an improved spread spectrum approach—the multi-switching frequency and multi-duty cycle (MFMD) scheme—in which the inductive-resonant WPT system can operate at multiple switching frequencies (e.g., 3 different frequencies) and as a part of a control signal with a specific switching frequency, there is a specific duty cycle. As it will be shown in the paper, the advantages of the proposed spread spectrum technique are: (1) due to the more even distribution of EME energy within the first-harmonic sidebands, it can give a better conducted EME reduction than conventional conducted EME reduction techniques; (2) a better efficiency of WPT system; (3) it is a simple approach because it can be implemented by using even an inexpensive 8-bit microcontroller, such as Atmega AVR 328p. Despite the fact that the proposed spread-spectrum approach is applied to a WPT system connected to a local 48-V DC grid, it can also be applied to a WPT system connected to AC grid to improve performance of the WPT system. The research is important because it contributes to the development of low-EME inductive-resonant WPT systems with improved performance characteristics.

The paper is organized as follows: the proposed spread spectrum approach is explained in detail in Section 2; the experimental setup is described in Section 3; the experimental results are presented and discussed in Section 4; and finally, conclusions are given in Section 5.

## 2. Proposed Improved Spread Spectrum Technique

### 2.1. Conventional Spread Spectrum Techniques

Historically, spread spectrum techniques were adapted from telecommunications engineering by power electronics specialists to reduce conducted and radiated EME in conventional non-resonant switching power converters in the 1990s. If the EME reduction technique is used, then the switching frequency harmonics in the EME spectrum are spread over a wider range of frequencies, leading to a significant reduction of peak EME levels. Recently, the spread spectrum techniques have also been applied to inductive-resonant WPT systems (and resonant wireless chargers) to reduce both radiated and conducted EME [10–13,15]. The conventional spread spectrum techniques mainly include periodic or random switching frequency modulation. Recently, a multi-switching frequency scheme was also proposed [13] to reduce EME in the WPT systems. In the periodic switching frequency modulation technique, the switching frequency is modulated by a modulating signal, such as a sine, triangle or sawtooth waveform. In the random modulation techniques, the switching frequency is modulated randomly by a random modulating signal with different probability distributions. In the multi-frequency technique, a WPT system operates at two or more different switching frequencies ($f_1, f_2, \ldots, f_k$) and the modulation period $T_m$ comprises $k$ time intervals ($\tau_k$) in which the WPT system operates at different frequencies $f_k$ (see Figure 3) [7]. More information about the technique can be found in [7] or [13].

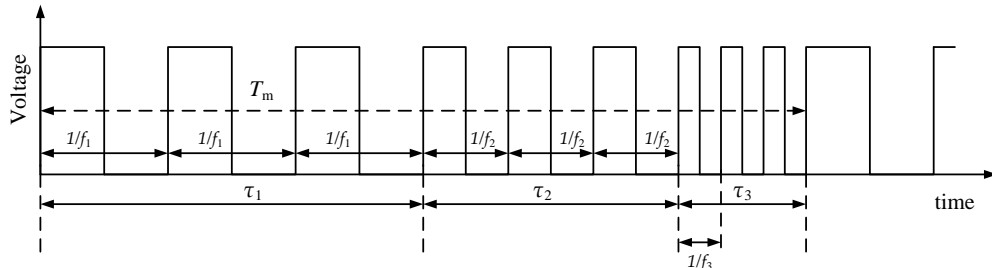

**Figure 3.** Waveform of a square signal with 3F technique [7]. During $\tau_1$ the system operates at frequency $f_1$, during $\tau_2$ the system operates at frequency $f_2$, and during $\tau_3$ the system operates at frequency $f_3$.

According to [15], the main disadvantages of random switching frequency modulation are the complexity and cost of implementation, because quite an expensive control block is applied based on a digital signal processor and a field-programmable gate array. As it is pointed out in [7], a microcontroller with very accurate timing is required to obtain the best performance from the periodic switching frequency modulation. In contrast to the periodic switching frequency modulation, the multi-switching frequency technique does not require a microcontroller with highly accurate timings, thus having only a slightly worse performance than periodic switching frequency modulation [13].

### 2.2. The Problem Description

To better understand the problems a designer will face when applying the conventional spread spectrum techniques, we resort to the simulations of the model of the inductive-resonant WPT system, which is depicted in Figure 4. The model can be used to calculate the conducted EME levels (remember that the conducted EME is the line impedance stabilization network (LISN) output voltage) and the efficiency for the WPT system without the spread spectrum, with the multi-frequency (3-frequency) technique and with the proposed technique. The model is assumed to be connected to a local DC 48 V grid. The model represents an inductive-resonant wireless battery charger operating in constant current (CC) or constant voltage (CV) modes. Typical values of the power components' parasitic resistances are taken into account to obtain more realistic values of the efficiency. However, switching losses in the power components are not considered in the model.

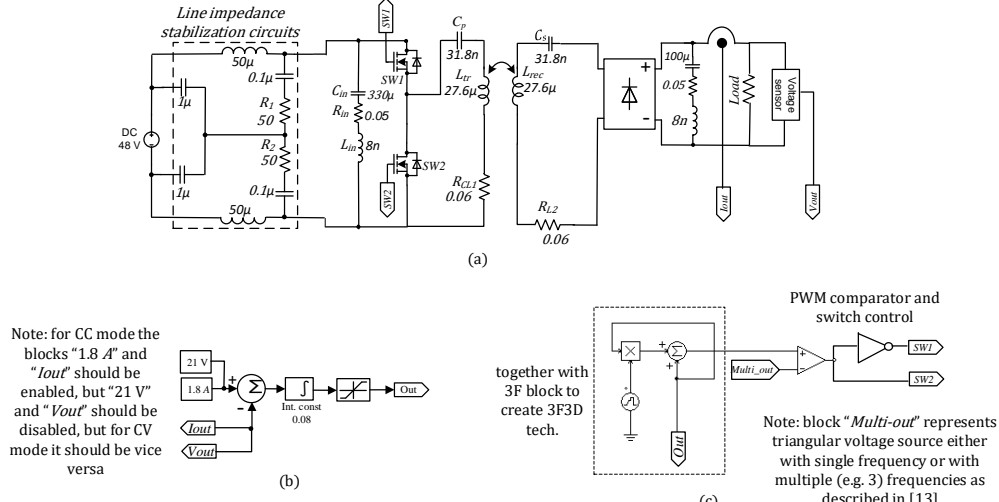

**Figure 4.** The model of the inductive-resonant WPT system under study: (**a**) power stage; (**b**) CC CV control; (**c**) PWM comparator and a block to create different duty cycles. Note: in CC mode output current is 1.8 A, but in CV mode output voltage is 21 V. The wireless charger can be used to charge 5-cell Li-ion batteries. Also note: the staircase voltage generator in should be zeroed if 3F3D technique is not used.

There are two main problems with the application of the conventional spread spectrum techniques: (1) a reduction in the efficiency; and (2) a significant distortion of the fundamental harmonic sidebands which leads to worse conducted EME reduction than is expected (see Figure 5). To better understand the reasons of the efficiency reduction and why the conducted EME levels do not reduce properly when applying the conventional spread spectrum techniques, let us analyze Figure 6 in which the input current and the efficiency versus frequency are shown.

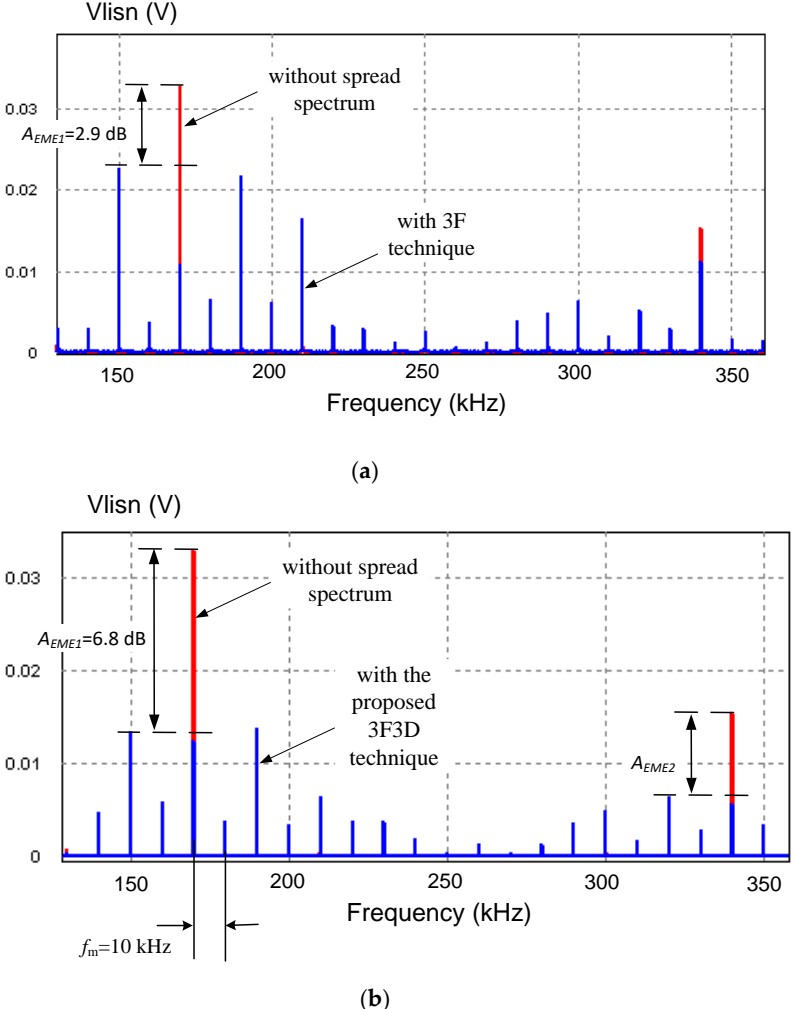

(**a**)

(**b**)

**Figure 5.** Simulated conducted EME (LISN output voltage) spectra: (**a**) when the spread spectrum is not used and when the conventional 3F (three-frequency) technique is used; (**b**) when the spread spectrum is not used and when the proposed 3F3D technique is used. Note: switching frequencies are 150, 170 and 190 kHz; for 3F3D technique $D_1$ = 20%; $D_2$ = 50%; and $D_3$ = 22%.

The efficiency decreases due to the fact that the efficiency of the inductive-resonant WPT system is a frequency-dependent quantity, as it can be seen in Figure 6. The maximum efficiency is usually at or in the vicinity of the resonant frequency. When using the conventional spread spectrum techniques, the switching frequency of the WPT system's inverter deviates from its central value, which is equal to the resonant frequency, and it results in a reduction in the efficiency.

The noticeable distortion of the conducted EME spectrum fundamental harmonic sidebands resulting in significantly reduced EME takes place due to the fact that the input current peak value is switching frequency-dependent. As it is seen in Figure 6, there are two maxima of the peak input current. If due to the spreading of the conducted EME spectrum, new spectrum components appear in the vicinity of the frequencies at which the

input current peak values are maximal, then a huge increase in conducted EME levels is observed close to those frequencies, leading to a decrease in the EME reduction.

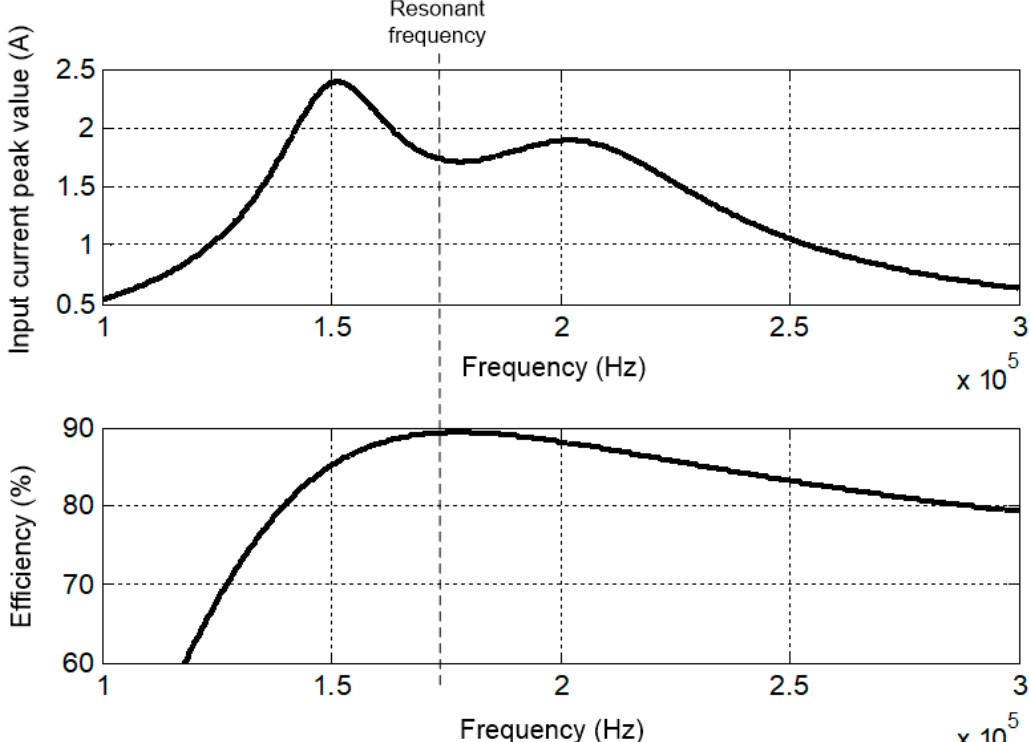

**Figure 6.** Theoretical input current peak value and the efficiency versus frequency of the inductive-resonant WPT system under study.

It is very important to note that the simulations show that better results in terms of conducted EME reduction can be achieved if the difference between maximum and minimum switching frequencies ($f_{3-1}$) is at least four times the modulation frequency $f_m = 1/T_m$ (see Figure 3). However, a higher $f_3 - f_1$ leads to a worse efficiency. Therefore, taking into account the tradeoff between the conducted EME reduction and the efficiency, it is better to choose $f_3 - f_1 = 4$ (both $f_1$ and $f_3$ should be within allowed WPT frequency range), but $f_m$ should be equal to 10 kHz, because it is well known from the spectrum analysis theory that $f_m$ should be chosen equal to or slightly higher than the resolution bandwidth (RBW) of a spectrum analyzer used in the measurements of the conducted EME [10]. Note that RBW is the bandwidth of the intermediate frequency filter of a spectrum analyzer, and according to CISPR 11, it should be equal to 9 kHz during the conducted EME measurements within 150 kHz–30 MHz.

### 2.3. Description of the Proposed Spread Spectrum Technique

To solve the aforementioned problems, we propose the usage of the multi-frequency, multi-duty cycle technique. In this technique, we can use reduced values of duty cycles $D_1$ and $D_3$ with respect to $D_2$ to reduce the amplitudes (and power) of the spectrum components, hence reducing peak conducted EME levels in the vicinity of the input current maxima frequencies and increasing efficiency (because the conduction losses are proportional to the amplitudes of the spectrum components of the power components' currents). We will compare the three-frequency three-duty cycle (3F3D) technique with the conventional three-frequency (3F) technique.

The inverter control signal waveform for the case of 3F3D technique application is depicted in Figure 7. In the proposed technique, there are three intervals ($\tau_1$, $\tau_2$ and $\tau_3$) at which the signal frequencies are $f_1, f_2$ and $f_3$, respectively. The difference in this technique when compared to the conventional 3-frequency technique is that duty cycles $D_1, D_2$ and

$D_3$ of the signal fragments during $\tau_1$, $\tau_2$ and $\tau_3$, respectively, are not equal. It is logical that to reduce peak EME levels at frequencies $f_1$ and $f_3$, $D_1$ and $D_3$ should be lower than $D_2$.

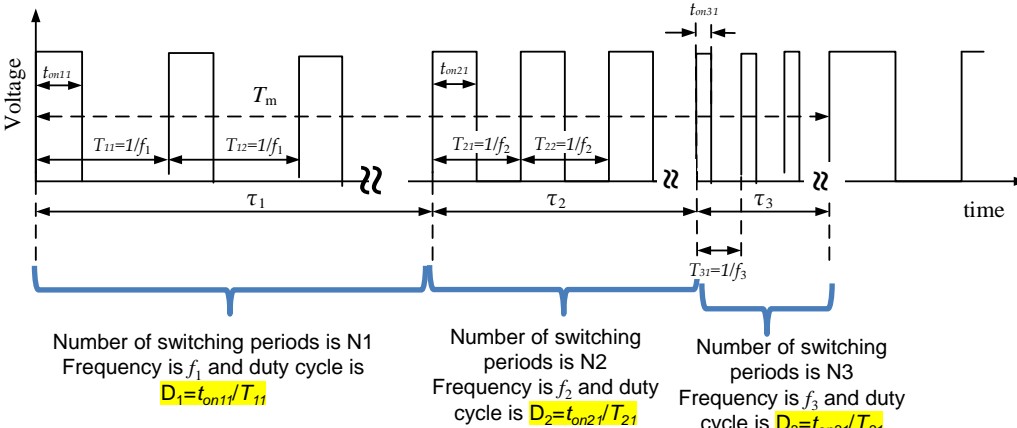

**Figure 7.** Simplified 3F3D waveform of the inverter control signal used in the experiments. If $N1 = N2 = N3 = 6$, $f_1 = 151.5$ kHz, $f_2 = 170.6$ kHz, $f_3 = 190.1$ kHz, then $f_\mathrm{m} = 10.6$ kHz.

Figure 5 clearly demonstrates the usefulness of the proposed 3F3D spread spectrum technique in the comparison to the conventional spread spectrum technique: the reduction coefficients ($A_{\mathrm{EME1}}$ and $A_{\mathrm{EME2}}$) of the first and the second harmonics peak levels are increased significantly. Moreover, the simulations also show that the proposed technique also leads to a moderate increase in efficiency.

### 3. Experimental Setup

In order to study the effect of the conventional spread-spectrum technique and MFMD spread spectrum technique (in our case it is 3F3D technique) on the conducted EME of the inductive-resonant WPT system, an experimental prototype of the system has been designed and practically built. A block diagram of the prototype is presented in Figure 8. It consists of a half-bridge inverter (from GaN systems), the half-bridge inverter dead time control circuit (homemade), a microcontroller (ATmega328PB Xplained Mini evaluation board), the primary-side and the secondary-side series compensation circuits (based on multiple capacitors), the self-made transmitting and receiving coils $L_1$ and $L_2$, respectively (made of suitable litz wire and a 10X10 cm ferrite pad), the secondary-side high-frequency full-wave rectifier with a high-frequency ripple filter (electrolytic and ceramic capacitors connected in parallel) and a simple input conducted EME filter (consisting of a 10 μF electrolytic capacitor connected in parallel to a 1 μF film capacitor). The dead time control circuit generates two square signals to control the inverter transistors adding some time delay to prevent the inverter's transistors from the shoot-through phenomenon. The half bridge inverter input is connected to the output of 48 V DC power supply. One more power supply is used to feed the control circuits and drivers, except for the microcontroller evaluation board, which is powered by the same laptop used for reprogramming the microcontroller. The primary and the secondary compensation capacitances were calculated so that the primary and secondary-side resonant tanks are tuned to the resonance (at the resonance frequency). The WPT system was designed for the operation within the Qi standard allowed range of frequencies (this is why the central switching frequency $f_0$ is 170.6 kHz).

The values of the main parameters of the designed inductive-resonant WPT system are shown in Table 1. The prototype can be used without spread spectrum, with 3F and the proposed 3F3D techniques.

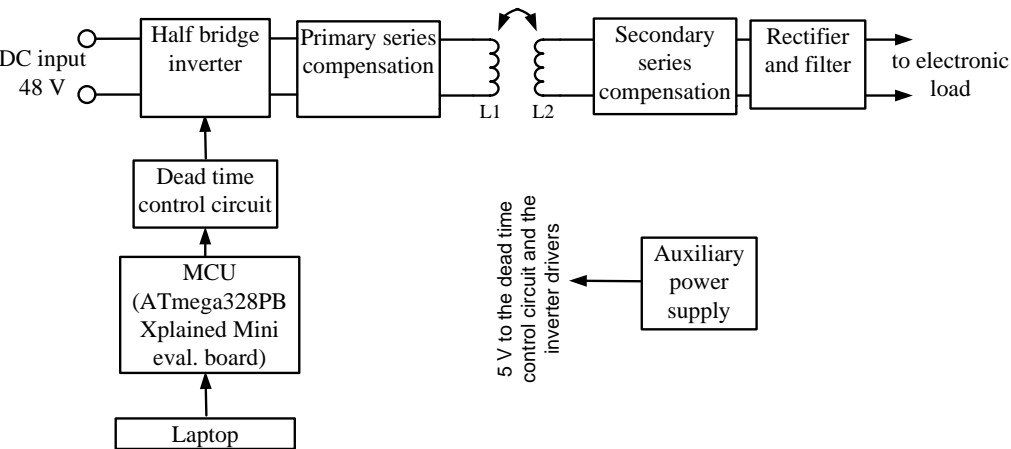

**Figure 8.** A block diagram of the experimental prototype.

**Table 1.** The parameters values of the designed WPT system.

| Parameter | Numerical Value | Unit of Measurement |
|---|---|---|
| Nominal output current in CC mode [1] | 1.8 | A |
| Nominal output voltage in CV mode [2] | 21 | V |
| Resonant frequency $f_{rez}$ | 170 | kHz |
| DC input voltage | 48 | V |
| Range of allowed switching frequencies | 100–200 | kHz |
| Primary inductance L1 and secondary inductance L2 | 26.2 | μH |
| Ferrite pad size | $10 \times 10$ | cm |
| Nominal distance between the coils | 2.8 | cm |
| Primary compensation capacitance | 31.79 | nF |
| Secondary compensation capacitance | 31.79 | nF |
| The worst-case coupling coefficient | 0.33 | - |
| Maximum misalignment of the coils | 1 | cm |
| Range of the load resistances | 10–117 | Ω |

[1] Note: the WPT system can operate in natural CC mode (when the load resistance is between 10 and 11.7 Ω), because series–series compensation is used. [2] Note: due to absence of the closed-loop control, the WPT system can operate in emulated CV mode (in range of load resistances between 11.7 and 117 Ω) by adjusting the control signal (from the output of the microcontroller) average duty cycle by using reprogramming the microcontroller for a given value of the load resistance.

Since the multi switching frequency technique (with 2–4 switching frequencies) or the proposed MFMD technique do not require accurate timings, a cheap low-resolution microcontroller can be used. Thus, for obtaining a square signal with multiple frequencies and duty cycles as shown in Figure 3 or Figure 7, we used the microcontroller ATmega328PB Xplained Mini evaluation board. For the generation of the pulse sequence, a direct microcontroller port manipulation (using *delay_us*) was used. By composing the microcontroller program code, modulation (repetition) frequency $f_m$ and the maximum difference between switching frequencies $\Delta f = f_3 - f_1$ can be simply achieved.

A photo of the experimental setup is shown in Figure 9. To measure the spectrum of the input conducted EME of the WPT system, a mixed domain digital oscilloscope Tektronix MDO4034B with separate spectrum analyzer as well as a homemade DC LISN (according to the circuit diagram presented in Figure 4) were used. When measuring the efficiency, the output power was measured by using the electronic load, but the input power was measured by using a suitable DC ammeter and a DC voltmeter. During the efficiency measurements, the LISN was disconnected from the circuit. To set different values of the load resistances, the electronic load in constant resistance mode was used.

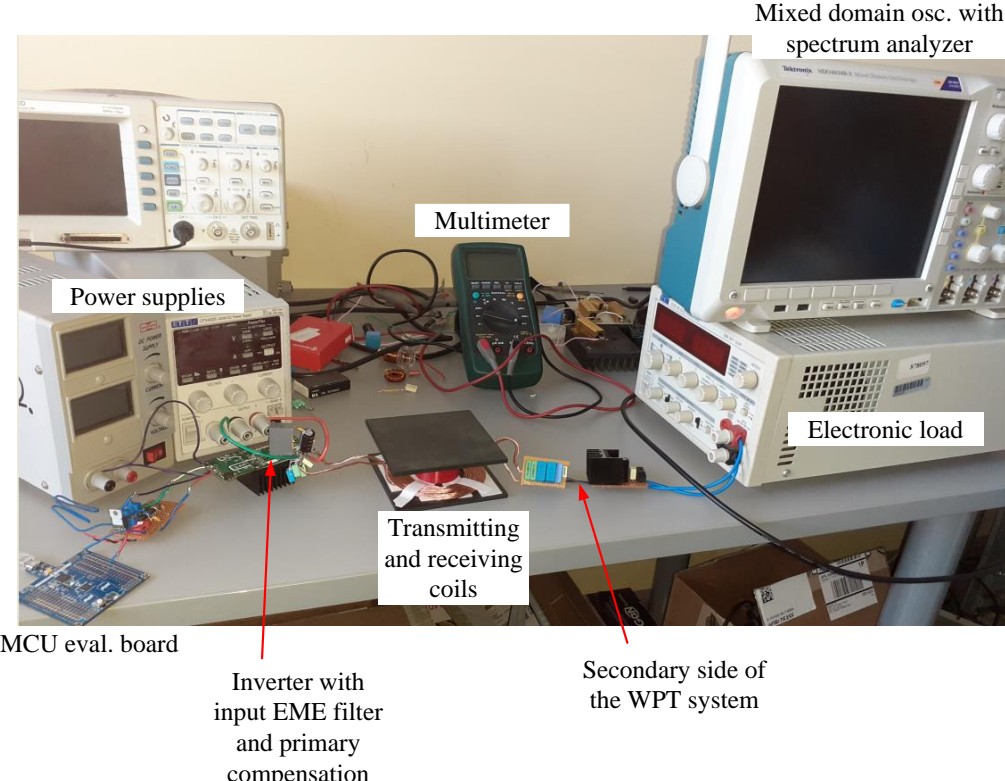

**Figure 9.** Image of the experimental setup. Note: DC LISN is not shown here, but it was used during the measurements of conducted EME.

## 4. Results and Discussion

The measurements of the conducted EME were made according to the CISPR 11 standard within the frequency range 150 kHz–30 MHz, with the RBW of the spectrum analyzer as 9 kHz (as required by the standard). The main measurements were made for a nominal distance between the coils (2.8 cm), maximum misalignment (1 cm) of the coils and at the load resistance 11.7 Ω (end of CC mode), because in this case the conducted EME levels are the highest. Some measurements were made also for other misalignments of the coils and other resistances of the load. A peak detector was used during the measurements. The obtained results were saved in CSV format files and then post-processed using Matlab. To obtain more accurate results, we used an average of 16 consecutive sweeps.

Despite the fact that the WPT system was examined in an open-loop mode, the average duty cycle of the control signal was adjusted to obtain the same value of output current (in CC mode) or output voltage (in CV mode) regardless the load resistance or the applied spread spectrum technique. This imitation of the closed-loop control made the comparable measurements under different control schemes possible, because when different spread spectrum techniques are applied, the output voltage or current can be changed.

The main measurement results are presented in Figures 10–12 and Table 2. For a reference purposes, CISPR 11 standard (quasi peak) limit lines for industrial (Class A) Group 1 devices connected to a DC grid are also depicted.

**Table 2.** The first and the second harmonic amplitude reduction coefficients ($A_{EME1}$ and $A_{EME2}$) and the efficiencies ($\eta$) for different cases.

| Case | $A_{EME1}$ (dB) | $A_{EME2}$ (dB) | $\eta$ (%) |
|---|---|---|---|
| No spread spectrum | - | - | 85.69 |
| Conventional 3F technique | 2.2 | 3.4 | 84.5 |
| Proposed 3F3D technique | 6.8 | 5.9 | 85.33 |

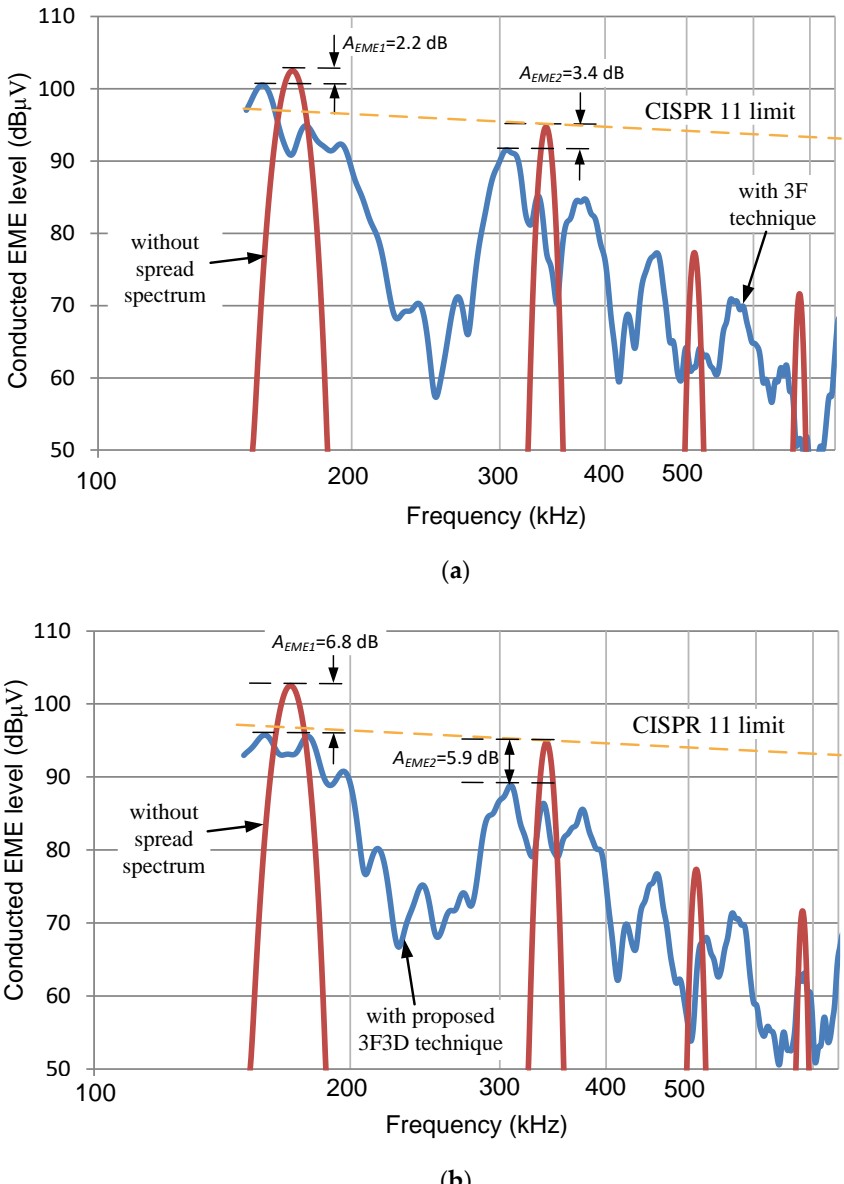

(a)

(b)

**Figure 10.** Comparison of spectra of the conducted EME (150–750 kHz) from the WPT system without the spread spectrum and with the spread spectrum based on 3-frequency technique (**a**) and based on the proposed 3F3D spread spectrum technique (**b**). Note: $f_m$ = 10.6 kHz, $f_1$ = 151.5 kHz, $f_2$ = 170.6 kHz, $f_3$ = 190.1 kHz, RBW = 9 kHz. Also note: (1) if the spread spectrum was not used, then $D$ = 34%; (2) if the 3-frequency technique was used, then $D_1 = D_2 = D_3$ = 34%; and (3) if the proposed 3F3D technique was used, then $D_1$ = 30%, $D_2$ = 50; $D_3$ = 30%. The results are shown for the worst-case peak EME levels.

In order to quantitatively assess the conducted emission reduction, the reduction coefficient of the amplitude of the fundamental harmonic of the conducted EME ($A_{EME1}$) and the reduction coefficient of the amplitude of the second harmonic of the conducted EME ($A_{EME2}$) were calculated from the measurements, because the harmonics are the most dominant ones (especially the fundamental harmonic). Note that the reduction coefficient of the amplitude of the harmonic is the difference (expressed in dB) between the amplitude of the harmonic of the WPT system without the spread spectrum and the maximum value of the amplitudes of the spectrum components of the given harmonic sidebands of the WPT system with the spread spectrum.

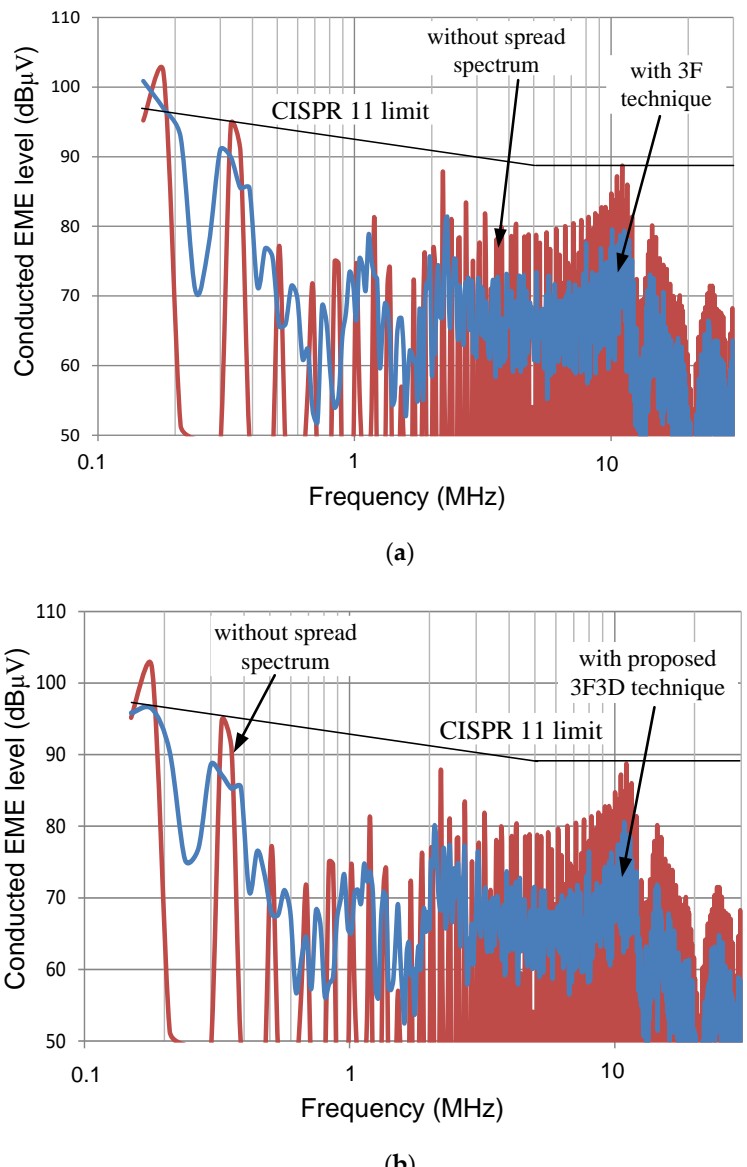

**Figure 11.** Comparison of spectra of the conducted EME (0.15–30 MHz) from the WPT system without the spread spectrum and with the spread spectrum based on 3-frequency technique (**a**) and based on the proposed 3F3D spread spectrum technique (**b**). Note: other parameters are the same as those in Figure 10.

Initially the WPT system was examined without the spread spectrum, then with 3-frequency spread spectrum technique and then with the proposed MFMD (3F3D) technique. In all cases, the duty cycles were experimentally determined and their values were added to the program code to obtain the same output DC voltage or current regardless of the control method. For the proposed 3F3D technique, the optimum values of the duty cycles D1, D2 and D3 (for the worst-case coupling coefficient and when the battery equivalent input resistance is at the boundary between CC and CV modes) were initially determined by using the simulations of the model shown in Figure 4 and then more accurate optimum values were determined experimentally and included in the program code.

As the measurements results show, the application of both spread spectrum techniques results in a significant reduction of conducted EME (at some frequency ranges up to 10 dB) along with a reduction in efficiency. The proposed 3F3D technique gives a considerably better conducted EME levels reduction than the conventional 3-frequency spread spectrum technique at a lower frequency range (up to 1.2 MHz). The conducted EME levels exceed

the CISPR 11 limit in the cases of both the 3-frequency technique and without the spread spectrum, but the proposed 3F3D technique results in significant improvements in conducted EME levels, leading to a shift of the conducted EME levels below the maximally allowable the CISPR 11 limit. Moreover, the proposed technique also moderately improves the efficiency when comparing the WPT system with the 3-frequency scheme (Table 2), and some experiments also show that the RMS values of the power components currents were decreased by several percent. All the aforementioned improvements come at no cost. All that required is to find the optimum values of the duty cycles, create a suitable code and copy it to the microcontroller memory. Note that the proposed technique also showed improvements for other resistances of the load and for different misalignments of the coils. Note that optimum values of D1 – D3 are dependent on the load resistance and the coupling coefficient, and they were determined only for the lowest coupling coefficient and the highest load resistance in CC mode (i.e. 11.7 $\Omega$). Therefore, the proposed spread spectrum technique may not give better EME reduction than that of the conventional spread spectrum technique at much higher load resistances (in CV mode, higher than several tens of $\Omega$), but this is not a problem because, as mentioned before, conducted EME is the highest when the coupling coefficient is the lowest and when the battery equivalent input resistance is at the boundary between CC and CV modes (in our case it is 11.7 $\Omega$).

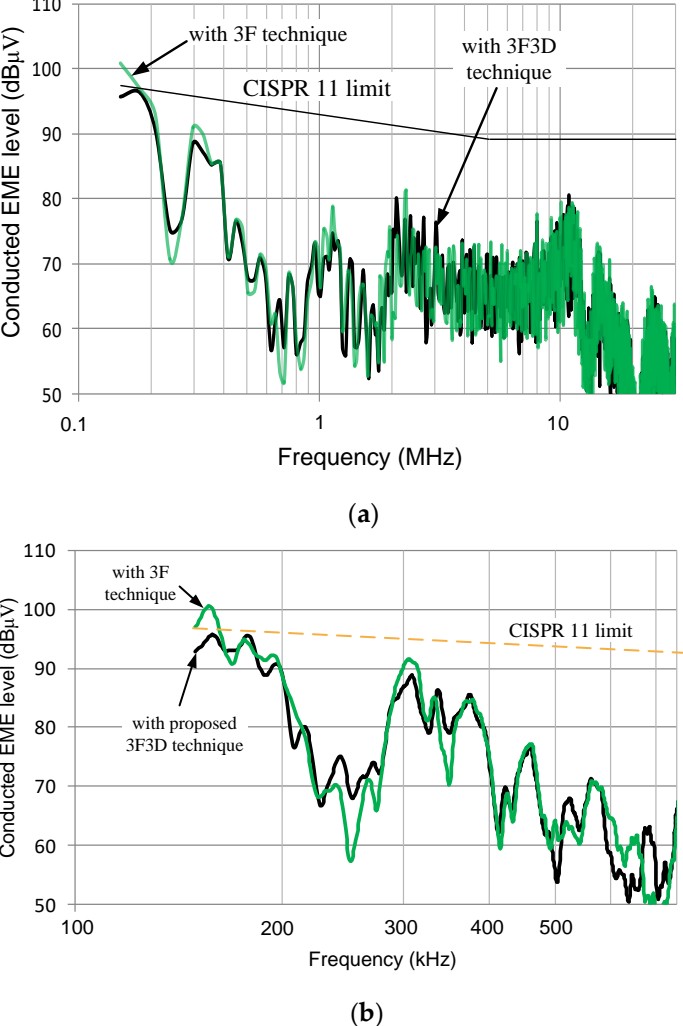

(**a**)

(**b**)

**Figure 12.** Comparison of spectra of the conducted EME from the WPT system with the spread spectrum based on 3-frequency technique and based on the proposed 3F3D spread spectrum technique: (**a**) within frequency range 0.15–30 MHz; and (**b**) within frequency range 0.15–0.8 MHz. Note: other parameters are the same as those in Figure 10.

## 5. Conclusions

The presented study shows that due to the frequency characteristics of the resonant WPT system, conventional spread spectrum techniques may not give a considerable reduction in conducted emissions, because of the significant increase in the conducted emissions levels at the frequencies close to the frequencies at which the input current is maximum. Moreover, due to the fact that the losses depend on the frequency, the efficiency goes down as switching frequency significantly deviates from the resonant frequency. In order to improve the reduction in conducted emissions, a novel spread spectrum technique—the three-frequency and three-duty cycle (3F3D) technique—was proposed. The advantages of the proposed spread spectrum technique are: (1) due to more even distribution of the conducted emissions energy within the first-harmonic sidebands, it can give a better reduction in conducted emissions than conventional conducted emission reduction techniques; (2) a better efficiency of WPT system; (3) it is a simple approach because it can be implemented by using an inexpensive 8-bit microcontroller. In the novel spread spectrum technique, a better conducted emission reduction and efficiency are achieved by choosing lower values of duty cycles of the control signal at minimum and maximum switching frequencies.

The study presented may be useful for engineers and researchers who develop wireless power transfer systems with reduced conducted electromagnetic emissions.

**Author Contributions:** Conceptualization, D.S.; methodology, D.S. and D.D.S.; software and design, D.D.S. and A.S.; validation, D.D.S. and D.S.; formal analysis, D.S. and D.D.S.; data curation, D.D.S. and A.S.; writing—original draft preparation, D.S.; writing—review and editing, J.Z.; visualization, J.Z.; supervision, J.Z.; project administration, J.Z. All authors have read and agreed to the published version of the manuscript.

**Funding:** This work has been supported by the European Regional Development Fund within the Activity 1.1.1.2 "Post-doctoral Research Aid" of the Specific Aid Objective 1.1.1: "To increase the research and innovative capacity of scientific institutions of Latvia and the ability to attract external financing, investing in human resources and infrastructure" of the Operational Program "Growth and Employment" (No.1.1.1.2/VIAA/3/19/415).

**Data Availability Statement:** Data of our study are available upon request.

**Acknowledgments:** We would like to express our gratitude to A. Zhiravecka for her help in improving our English grammar.

**Conflicts of Interest:** The authors declare no conflict of interest.

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
