# Peer review of "An Improved Spread-Spectrum Technique for Reduction of Electromagnetic Emissions of Wireless Power Transfer Systems"

_electronics, doi:10.3390/electronics11172733_

Round 1

Reviewer 1 Report

The paper is written and organized well. I have following points which may improve the work.

1. Comment Only: Figures in the paper have references, it is assumed that authors’ have the reproduction rights of those figures.

2. Organization of the paper shall be added in the end of Section 1.

3. In Section 2.3, it is suggested to include a flowchart or a pseudocode of the operation.

4. In results, can you place 3f and 3f3d in a new albeit single figure for analysis and comparison.

5. Writing part:

i. Line 14: do you mean to say ‘inexpensive’ as compared to ‘cheap’. Also, in line 81, wording needs to be rechecked.

ii. Sentence in Line 27-28 needs rephrasing for clarity.

iii. Acronym styles shall be consistent, e.g., in line change ‘ISM (industrial, scientific and medicine)’ to  ‘industrial, scientific and medicine (ISM)’.

iv. Line 92-95 may be moved to a footnote in the paper, and the reference of the thesis be included in the references section.

v. Conclusion section needs to be better written.    

Author Response

Thank you very much for your valuable comments and suggestions. All the corrections are shown in yellow in the manuscript.

  1. Comment Only: Figures in the paper have references, it is assumed that authors’ have the reproduction rights of those figures.

Both figures 2 and 3 are not infringement of copyrights, because figure 2 is copied from an open source internet site https://www.academyofemc.com/emc-standards, then it was modified by 50% and official permission of the copyright holder (The Academy of EMC) is obtained, but figure 3 (prepared by me) is copied from my MDPI open source journal paper.

  1. Organization of the paper shall be added in the end of Section 1.

Thank you very much for your suggestion. We added a sentence at the end of Section 1. The paper is organized as follows: the proposed spread spectrum approach is explained in details in Section 2; the experimental setup is described in Section 3; the experimental results are presented and discussed in Section 4, and finally, conclusions are given in Section 5.

  1. In Section 2.3, it is suggested to include a flowchart or a pseudocode of the operation.

We did not include program codes for the microcontroller, because the spread spectrum techniques can be implemented by using different approaches (and codes) and the codes are relatively long. So, we think that it is redundant to add the codes. To better understand how the spread spectrum techniques were implemented with the microcontroller in Section 3 we put a sentence “For generation of the pulse sequence, a direct microcontroller port manipulation (using delay_us) was used.”

Complete program code to generate pulse sequence with the microcontroller is shown below:

#define F_CPU 15512612UL

#include <avr/io.h>

#include <avr/delay.h>

#include <util/delay.h>

int main(void)

{

                  DDRB = 0xFF;           // Output

                  PORTB = 0x00;          // Clear the PORTB

                 while (1)

                  {

                                    _delay_us(4.522);

                                    PORTB |= (1 << PINB2);                               // Set

                                    _delay_us(1.938);

                                    PORTB &= ~(1 << PINB2);                          // Reset

                                    _delay_us(4.522);

                                    PORTB |= (1 << PINB2);                               // Set

                                    _delay_us(1.938);

                                    PORTB &= ~(1 << PINB2);                          // Reset

                                    _delay_us(4.522);

                                    PORTB |= (1 << PINB2);                               // Set

                                    _delay_us(1.938);

                                    PORTB &= ~(1 << PINB2);                          // Reset

                                    _delay_us(4.522);

                                    PORTB |= (1 << PINB2);                               // Set

                                    _delay_us(1.938);

                                    PORTB &= ~(1 << PINB2);                          // Reset

                                    _delay_us(4.522);

                                    PORTB |= (1 << PINB2);                               // Set

                                    _delay_us(1.938);

                                    PORTB &= ~(1 << PINB2);                          // Reset

                                    _delay_us(4.522);

                                    PORTB |= (1 << PINB2);                               // Set

                                    _delay_us(1.938);

                                    PORTB &= ~(1 << PINB2);                          // Reset

                                    _delay_us(2.85);

                                    PORTB |= (1 << PINB2);                               // Set

                                    _delay_us(2.85);

                                    PORTB &= ~(1 << PINB2);                          // Reset

                                    _delay_us(2.85);

                                    PORTB |= (1 << PINB2);                               // Set

                                    _delay_us(2.85);

                                    PORTB &= ~(1 << PINB2);                          // Reset

                                    _delay_us(2.85);

                                    PORTB |= (1 << PINB2);                               // Set

                                    _delay_us(2.85);

                                    PORTB &= ~(1 << PINB2);                          // Reset

                                    _delay_us(2.85);

                                    PORTB |= (1 << PINB2);                               // Set

                                    _delay_us(2.85);

                                    PORTB &= ~(1 << PINB2);                          // Reset

                                    _delay_us(2.85);

                                    PORTB |= (1 << PINB2);                               // Set

                                    _delay_us(2.85);

                                    PORTB &= ~(1 << PINB2);                          // Reset

                                    _delay_us(2.85);

                                    PORTB |= (1 << PINB2);                               // Set

                                    _delay_us(2.85);

                                    PORTB &= ~(1 << PINB2);                          // Reset

                                    _delay_us(3.5);

                                    PORTB |= (1 << PINB2);                               // Set

                                    _delay_us(1.6);

                                    PORTB &= ~(1 << PINB2);                          // Reset

                                    _delay_us(3.5);

                                    PORTB |= (1 << PINB2);                               // Set

                                    _delay_us(1.6);

                                    PORTB &= ~(1 << PINB2);                          // Reset

                                    _delay_us(3.5);

                                    PORTB |= (1 << PINB2);                               // Set

                                    _delay_us(1.6);

                                    PORTB &= ~(1 << PINB2);                          // Reset

                                    _delay_us(3.5);

                                    PORTB |= (1 << PINB2);                               // Set

                                    _delay_us(1.6);

                                    PORTB &= ~(1 << PINB2);                          // Reset

                                    _delay_us(3.5);

                                    PORTB |= (1 << PINB2);                               // Set

                                    _delay_us(1.6);

                                    PORTB &= ~(1 << PINB2);                          // Reset

                                    _delay_us(3.5);

                                    PORTB |= (1 << PINB2);                               // Set

                                    _delay_us(1.6);

                                    PORTB &= ~(1 << PINB2);                          // Reset

               }

}

  1. In results, can you place 3f and 3f3d in a new albeit single figure for analysis and comparison.

Thank you very much for the suggestion. Spectra of conducted EME if conventional 3f and the proposed 3f3d techniques are used are compared in Fig. 12.

  1. Writing part:

i. Line 14: do you mean to say ‘inexpensive’ as compared to ‘cheap’. Also, in line 81, wording needs to be rechecked.

Yes, it is better to say “inexpensive”. We made also some modifications in line 81.

Sentence in Line 27-28 needs rephrasing for clarity.

We rephrased the sentence for more clarity: “With an enormous development of different electronic and electrical devices wireless power transfer (WPT) has become a very popular topic of research nowadays, because WPT is more reliable and convenient approach for power transmission than the traditional power transmission with wires”.

iii. Acronym styles shall be consistent, e.g., in line change ‘ISM (industrial, scientific and medicine)’ to ‘industrial, scientific and medicine (ISM)’.

Thank you very much for the suggestion. We have now written “industrial, scientific and medicine (ISM)”.

Line 92-95 may be moved to a footnote in the paper, and the reference of the thesis be included in the references section.

We put “This paper is partly based on the results of a master thesis [15]” in footnote of page 2.

Conclusion section needs to be better written.

The conclusions are now better written.

Reviewer 2 Report

The authors proposed an improved spread spectrum approach –multi switching frequency and multi duty cycle (MFMD) scheme. In this approach, the inductive-resonant WPT system can operate at multiple switching frequencies and for a part of a control signal with specific switching frequency there is specific duty cycle. The technique can be simply implemented even using cheap 8-bit microcontroller.  The results look encouraging and motivating. But there are still some contents, which need be revised in order to meet the requirements of publish. A number of concerns listed as follows:

(1)   The abstract does not provide significant information and it should be revised to highlight the significant methodological contributions and conclusions.

(2)   In the introduction, it is necessary to add a research background introduction and a detailed explanation of the research motivation, so as to attract more potential audiences.

(3)   The methodology is not clear and it can be further improved it is better to add a flow chart of methodology.

(4)   Those related works and their relevance are required to analyze. You must add and review all significant similar works that have been done. For example, 10.3390/agriculture12060793; 10.1109/JSTARS.2021.3059451 ;10.1016/j.engappai.2022.105139 10.1007/s10489-022-03719-6 and so on.

(5)   In the introduction section, you should give the novelty and the contributions of your works.

(6)   The method/approach in the context of the proposed work should be written in detail.

(7)   Line 191, there is a grammatical error in this sentence. Please revise it. In addition, please proof read from native speaker.

(8)   Line 255, how to set these parameters values? The authors should provide the reason.

(9)   In the Conclusions, what are the advantages and disadvantages of this study compared to the existing studies in this area?

Author Response

Thank you very much for your valuable comments and suggestions. All the corrections are shown in green in the manuscript.

(1)   The abstract does not provide significant information and it should be revised to highlight the significant methodological contributions and conclusions.

Thank you very much for your comment. We added and rephrased some sentences in the abstract to improve it.

(2)   In the introduction, it is necessary to add a research background introduction and a detailed explanation of the research motivation, so as to attract more potential audiences.

Thank you very much for your comments. We improved the introduction, by explaining in more details the research motivation and novelty as shown in green on pages 2 and 3.

(3) The methodology is not clear and it can be further improved it is better to add a flow chart of methodology.

Thank you very much for your comment, but actually we should tell you that we did not propose new methodology. We proposed a novel spread spectrum technique (method) thanks to which the conducted EME reduction coefficient and the efficiency can be improved by reducing the emissions energy in the vicinity of input current peaks. The method is described relatively clearly in Section 2.

(4) Those related works and their relevance are required to analyze. You must add and review all significant similar works that have been done. For example, 10.3390/agriculture12060793; 10.1109/JSTARS.2021.3059451; 10.1016/j.engappai.2022.105139; 10.1007/s10489-022-03719-6 and so on.

We tried to read the aforementioned papers, but we concluded that they are not related to our topic – electromagnetic compatibility (EMC) issues in wireless power transfer (WPT) systems. Therefore, we did not cite them in our paper. But we analyzed related works [10] – [14] and cited them in our paper.

(5)   In the introduction section, you should give the novelty and the contributions of your works.

Thank you very much for your suggestion. Therefore, novelty of this paper is to use an improved spread spectrum approach – multi switching frequency and multi duty cycle (MFMD) scheme, in which the inductive-resonant WPT system can operate at multiple switching frequencies (e.g. 3 different frequencies) and for a part of a control signal with specific switching frequency there is a specific duty cycle. As it will be shown in the paper, the advantages of the proposed spread spectrum technique are: 1) due to more even distribution of EME energy within the first-harmonic sideband it can give better conducted EME reduction than that of conventional conducted EME reduction techniques; 2) better efficiency of WPT system; 3) it is a simple approach because it can be implemented by using even an inexpensive 8-bit microcontroller such as Atmega AVR 328p.

(6)   The method/approach in the context of the proposed work should be written in detail.

Thank you very much for your comment. Description of the method is given in Section 2.3.

(7)   Line 191, there is a grammatical error in this sentence. Please revise it. In addition, please proof read from native speaker.

The manuscript English language quality was improved by an English language teacher. Moreover, after the paper is accepted, it will be proof read by the MDPI language center before publication.

(8) Line 255, how to set these parameters values? The authors should provide the reason.

To set different values of the load resistances, the electronic load in constant resistance mode was used.

So, CC mode will be if the load resistance is in range between 10 and 11.7 ohms. Why? The reason is as follows: there are 5-cell Li-ion batteries with charge cut-off voltage 5 times 4.2 V (=21 V). The charge cut-off voltage is the voltage at which the battery is being charged in CV mode. The discharge cut-off voltage (voltage below which discharging is not allowed) of such batteries is 18 V. In this case the lowest value of the battery equivalent load resistance at which CC mode starts will be 18/1.8=10 ohms, but the highest value of the battery equivalent load resistance at which CC mode ends will be 21/1.8=11.7 ohms.  

(9)  In the Conclusions, what are the advantages and disadvantages of this study compared to the existing studies in this area?

Advantages of this study are that for the first time a novel spread spectrum technique called multi-frequency and multi-duty-cycle technique is proposed and analyzed. Advantages of the proposed spread spectrum technique are: 1) due to more even distribution of EME energy within the first-harmonic sideband it can give better conducted EME reduction than that of conventional conducted EME reduction techniques; 2) better efficiency of WPT system; 3) it is a simple approach because it can be implemented by using even an inexpensive 8-bit microcontroller such as Atmega AVR 328p. The conclusions are now rewritten.

Round 2

Reviewer 2 Report

According to the revised paper, I have appreciated the deep revision of the contents and the present form of this manuscript.  There is little content, which need be revised according to the comment of reviewer in order to meet the requirements of publish. A number of concerns listed as follows:

(1) The authors need to interpret the meanings of the variables.

(2) Please highlight your contributions in introduction.

(3) How to determine these parameters? The author should give a detailed explanation.

(4) The theoretical background of the proposed method is adequately detailed in the paper.

(5) The inspiration of your work must further be highlighted. Some suggested recent literatures should add in the revised paper.

(6) Further correct typological mistakes and mathematical errors.

Author Response

Thank you very much for your comments. All the changes are shown in yellow.

(1) The authors need to interpret the meanings of the variables.

 The meanings of the variables (Tm, fm, D1, D2, D3, AEME1, AEME2, RBW, L1 and L2) are now interpreted in the revised version (see page 5, Figure 7, Table 1, page 10, caption of Table 2). 

(2) Please highlight your contributions in introduction.

 Thank you very much for your comment. We added a sentence about the contributions:  The importance of the research work accomplished contributes to the development of low-EME inductive-resonant WPT systems with improved performance characteristics.

(3) How to determine these parameters? The author should give a detailed explanation.

Thank you very much for your question.

Determination of the modulation frequency is described in lines 180 – 183: fm should be equal to 10 kHz, because it is well known from the spectrum analysis theory that fm should be chosen equal to or slightly higher than the resolution bandwidth (RBW) of a spectrum analyzer used in the measurements of the conducted EME.

Choice of RBW is discussed in line 185.

For the proposed 3F3D technique the optimum values of the duty cycles D1, D2 and D3 were initially determined by using the simulations of the model shown in Figure 4 and then more accurate optimum values were determined experimentally and included in the program code (line 325 - 329).  

Determination of minimum (f1) and maximum (f3) frequencies is discussed in line 179 – 180: it is better to choose f3-f1=4 (both f1 and f3 should be within allowed WPT frequency range).

(4) The theoretical background of the proposed method is adequately detailed in the paper.

 Thank you very for your comment.

 (5) The inspiration of your work must further be highlighted. Some suggested recent literatures should add in the revised paper.

 We have been inspired by the research works [10] – [14].

We have already written the following: There are different papers related to the suppression of conducted EME using spread spectrum approach in the inductive-resonant WPT systems [10] – [13] which show that the spread spectrum technique along with moderate conducted EME reduction can lead to some “adverse effects” mainly in terms of decrease in efficiency. Moreover, as pointed out in [13], the conducted EME reduction coefficient is not high because of uneven distribution of the conducted EME energy within the first harmonic sideband especially for higher switching frequency deviations. Thus, it is of importance to propose an improved spread spectrum approach that can give a noticeably better conducted EME reduction along with better efficiency.

The research works suggested by the reviewer in the previous review unfortunately were not related to our paper topic EMC issues in WPT systems.

(6) Further correct typological mistakes and mathematical errors.

Thank you very for your comment. We tried to correct typological mistakes and mathematical errors.

Round 3

Reviewer 2 Report

This paper can be accepted now.